# Antiviral Activities of Carbazole Derivatives against Porcine Epidemic Diarrhea Virus In Vitro

**DOI:** 10.3390/v13122527

**Published:** 2021-12-16

**Authors:** Zheng Chen, Jinfeng Chen, Xiaodong Wei, Huiying Hua, Ruiming Hu, Nengshui Ding, Jinhua Zhang, Deping Song, Yu Ye, Yuxin Tang, Zhen Ding, Shaoyong Ke

**Affiliations:** 1Department of Preventive Veterinary Medicine, College of Animal Science and Technology, Jiangxi Agricultural University, Nanchang 330045, China; chenzheng@jxau.edu.cn (Z.C.); cjf15807957490@sina.com (J.C.); wxd123kwl@sina.com (X.W.); ying18336899431@sina.com (H.H.); Zhangjh1122@jxau.edu.cn (J.Z.); sdp8701@jxau.edu.cn (D.S.); yy@jxau.edu.com (Y.Y.); tang53ster@gmail.com (Y.T.); 2Jiangxi Engineering Research Center for Animal Health Products, College of Animal Science and Technology, Jiangxi Agricultural University, Nanchang 330045, China; 3Jiangxi Provincial Key Laboratory for Animal Health, Institute of Animal Population Health, College of Animal Science and Technology, Jiangxi Agricultural University, Nanchang 330045, China; hrmvet19@163.com; 4State Key Laboratory for Pig Genetic Improvement and Production Technology, Jiangxi Agricultural University, Nanchang 330045, China; dingyd2005@hotmail.com; 5Hubei Biopesticide Engineering Research Centre, Hubei Academy of Agricultural Sciences, Wuhan 430064, China; keshaoyong@163.com

**Keywords:** antiviral drug, carbazole derivatives, porcine epidemic diarrhea virus

## Abstract

Porcine epidemic diarrhea virus (PEDV), an enteric coronavirus, causes neonatal pig acute gastrointestinal infection with a characterization of severe diarrhea, vomiting, high morbidity, and high mortality, resulting in tremendous damages to the swine industry. Neither specific antiviral drugs nor effective vaccines are available, posing a high priority to screen antiviral drugs. The aim of this study is to investigate anti-PEDV effects of carbazole alkaloid derivatives. Eighteen carbazole derivatives (No.1 to No.18) were synthesized, and No.5, No.7, and No.18 were identified to markedly reduce the replication of enhanced green fluorescent protein (EGFP) inserted-PEDV, and the mRNA level of PEDV N. Flow cytometry assay, coupled with CCK8 assay, confirmed No.7 and No.18 carbazole derivatives displayed high inhibition effects with low cell toxicity. Furthermore, time course analysis indicated No.7 and No.18 carbazole derivatives exerted inhibition at the early stage of the viral life cycle. Collectively, the analysis underlines the benefit of carbazole derivatives as potential inhibitors of PEDV, and provides candidates for the development of novel therapeutic agents.

## 1. Introduction

Porcine epidemic diarrhea virus (PEDV) is an enveloped, single-stranded positive-sense RNA virus that belongs to genera *alphacoronavirus*, the family *Coronaviridae*. The genome of PEDV is approximately 28 kb in length, and can encode sixteen nonstructural proteins (nsp1-nsp16), an accessory protein ORF3 and four structural proteins, including spike (S), envelope (E), membrane (M), and nucleocapsid (N) proteins [1,2]. PEDV can infect swine of all ages, with clinical symptoms including anorexia, vomiting, watery diarrhea, dehydration, and high mortality [3,4]. Since the re-emergence of highly virulent strains and world-wide spread, leading to serious economic loss in the swine industry, PEDV has drawn much attention [5,6].

Regardless of the genogroup, PEDV can be deadly in pig populations with no immunity or low immunity, or a healthy status [7]. At present, enhanced biosecurity and immunoprophylaxis is a preventive and control strategy [7,8]. The immunization of pregnant sows is important in the control of epidemic PEDV, and in reducing the number of deaths in suckling piglets [7,9,10]. The most common practice was used to initiate herd immunity in US pig farms during the 2013–2017 epidemic when no PEDV vaccines were available [7,11]. However, the current feedback and vaccination protocols are frequently inefficient or unsafe, due to: no standardized protocol for feedback; poor capacities of current vaccines (live or killed) to induce lactogenic immunity; antigenic differences between vaccines with epidemic strains; possibility of continuous re-infection by PEDV used for feedback within herds; current live vaccines may revert to virulent PEDV or recombine with field PEDV strains to generate new strains after they are applied in the field [7,11,12,13,14,15]. Therefore, there is an urgent need to find new methods to prevent and control PEDV.

Naturally existing compounds have properties of chemical diversity and antiviral activity, and thus may have utility as therapeutic agents against coronaviral infections [16]. Among natural products, carbazole alkaloids are of an important class, belonging to nitrogen-containing aromatic heterocyclic compounds, and mainly distributed in coal tar and Rutaceae plants [17]. Due to the promising pharmacological activities and structural diversity, carbazole alkaloids have attracted high interest for their various biological properties, ranging from antimicrobial, antiprotozoal, insecticidal, and antiviral activities [18]. Natural and synthetic carbazole alkaloids been reported to exhibit inhibitory activities against human immunodeficiency virus (HIV) [19,20,21], hepatitis C virus (HCV) [22], coxsackie virus (CV) [23], herpes simplex virus (HSV), etc. [24,25].

In this study, we generated 18 carbazole derivatives, and identified No.5, No.7, and No.18 as anti-PEDV drugs. No.7 and No.18 were of low toxicity, and exhibited anti-PEDV function in a dose-dependent manner. Both No.7 and No.18 could inhibit PEDV infection at the stage of viral attachment. All together, we validated No.7 and No.18 of carbazole derivatives as potential and theoretical therapeutic drugs for PEDV.

## 2. Materials and Methods

### 2.1. Cells, Virus, and Regents

Vero-81 cells (African green monkey kidney cells) were purchased from the American Type Culture Collection (ATCC), and cultured in DMEM (Gibco, Waltham, MA, USA) containing 10% FBS and 1% penicillin–streptomycin at 37 °C, 5% CO_2_. An enhanced green fluorescent protein (EGFP) gene chimeric PEDV (EGFP-PEDV) (GenBank accession no. JQ023162.1) strain was manipulated and presented by Li Zhen from Shanghai Veterinary Research Institute [26]. Wild-type PEDV (CH/JXJA/2017) was isolated from a diarrhea pig farm [27]. Antibodies against PEDV N were generated in our lab [28]. Antibodies against glyceraldehyde-3-phosphate dehydrogenase (GAPDH) were purchased from Proteintech Group, Inc. (Wuhan, China). Carbazole derivatives were designed and synthesized in our lab, and dissolved in dimethyl sulfoxide (DMSO).

### 2.2. Fluorescence Microscope Observation

To investigate the antiviral effect of 18 carbazole derivatives on PEDV replication, vero-81 cells grown on 24-well-plates were incubated with 18 carbazole derivatives diluted in the cell culture medium at concentration of 10 μM/mL for 1 h (h). Then, cells were infected EGFP-PEDV for 2 h, subsequently incubated with the same concentration of 18 carbazole derivatives for 24 h, and the lights of EGFP under fluorescence microscope were observed.

### 2.3. RNA Isolation and Quantitative Real-Time PCR (qRT-PCR)

RNA was extracted from PEDV-infected or non-infected vero-81 cells using TRIzol reagent (TaKaRa, Dalian, China), and reversed into cDNA by using Reverse Transcriptase M-MLV (TaKaRa, Dalian, China) according to the manufacturer’s instructions. QRT-PCR was performed using AceQ Universal SYBR qPCR Master Mix (Vazyme, Nanjing, China) with the specific primers which are shown in Table 1. Data were normalized against GAPDH expression, and were expressed as fold differences between control and treated cells using the 2^−ΔΔCT^ method. The individual mRNAs in each sample were assessed three times independently.

### 2.4. Flow Cytometry Assay

Vero-81 cells were treated with No.5, No.7, and No.18 carbazole derivatives dissolved in a cell culture medium for 1 h, then infected with EGFP-PEDV for 2 h. After that, the cells were retreated with the same concentration of No.5, No.7, or No.18 carbazole derivatives for 24 h. The cells (1 × 10^6^) were harvested by trypsinization, washed and pelleted with phosphate balanced solution (PBS), and fixed using ice cold 70% ethanol at 4 °C for 1 h. Pelleted cells were then washed with PBS, and incubated in RNaseA (100 mg per mL) in PBS at 37 °C for 30 min. Propidium iodide (PI) (Sigma, Saint Louis, MO, USA) was added for 15 min at room temperature, and cells were then analyzed by flow cytometry. Each measurement was performed in triplicate.

### 2.5. Cytotoxicity Assay

Cells were seeded into 96-well-plates, and grown to 80% confluence. After washing three times with PBS, the cells were treated with increasing concentrations of No.5, No.7, and No.18 carbazole derivatives ranging from 0 to 80 μM. Mock-treated cells served as a control. After 24 h, the cells were washed with PBS, and the cytotoxicity of the drug in vitro was assessed using the Cell Counting Kit-8 (CCK8) (Beyotime, Shanghai, China) according to the manufacturer’s instructions. Absorbance was measured with a microplate reader (Thermo Fisher Scientific) at 450 nm. Cytotoxicity was expressed according to the following formula: Cytotoxicity (%) = {(Abs sample) − (Abs blank)}/{(Abs negative control) − (Abs blank)} × 100%.

### 2.6. Viral Titration Assay

To investigate the inhibitory effect of the drug on PEDV infection, vero-81 cells were seeded into 24-well-plates, and incubated with increasing concentrates of carbazole derivatives (10, 20, 40 μM for No.7; and 20, 40, 60 μM for No.18) for 1 h prior to viral inoculation. After that, cells were washed with PBS, and infected with PEDV for 2 h. The medium was changed with the same concentrates of carbazole derivatives used above for 24 h. Supernatants were collected for TCID_50_ to determine viral titer. Briefly, supernatant samples with a tenfold dilution from 10^−1^ to 10^−10^ were added into vero-81 cells in 96-well-plates for 3 days. Each dilution was added to eight wells. The cytopathic effect (CPE) was visualized via light microscopy, and TCID_50_ was calculated using the Reed–Muench [29,30].

### 2.7. Western Blot Analysis

Protein lysates were obtained from vero-81 cells using ice-cold cell lysis buffer containing 10 mM phenylmethylsulfonyl fluoride (PMSF). Equal volumes of samples were separated by 12% sodium dodecyl sulfate-polyacrylamide gel electrophoresis (SDS-PAGE), and then proteins were electroblotted onto a polyvinylidene fluoride (PVDF) membrane (Millipore, Waltham, MA, USA). Membranes were blocked with 10% nonfat milk for 2 h at 37 °C, and then incubated with primary antibody at 4 °C overnight. The following primary antibodies were used: anti-GAPDH or anti-PEDV N polyclonal antibodies. After that, horseradish peroxidase (HRP)-conjugated secondary antibodies (TransGen Biotech, Beijing, China) were incubated for 1 h. Protein bands were detected by enhanced chemiluminescence reagents (Beyotime, Beijing, China), and analyzed by ImageLab software.

### 2.8. Viral Attachment and Entry Assays

For viral attachment assay: vero-81 cells were seeded into 12-cell-plates, and grown to 80%. Non-toxic concentrations of No.7 (40 μM) or No.18 (40 μM) were mixed with PEDV (MOI = 0.1), and then added into the cells for 1 h at 4 °C. The cells infected with PEDV without drug treatment were set as control. After washing with PBS, the levels of viral RNA (vRNA) in the cells were determined by qTR-PCR.

For viral entry assay: vero-81 cells were infected with PEDV (MOI = 0.1) at 4 °C for 1 h. Unbound viruses were removed by washing with the maintenance medium. The cells were incubated with No.7 (40 μM) or No.18 (40 μM) at 37 °C for 2 h. As a control, another set of cells were infected with the same dose of PEDV without DMSO treatment. After removing the unbound viruses by washing with maintenance medium, the levels of vRNA in cells were determined by qRT-PCR.

### 2.9. Statistical Analyses

Results were expressed as the means ± standard deviation (SD) from three independent experiments. All statistical analyses were performed using GraphPad Prism 8.0 (GraphPad, San Diego, CA, USA). Values of *p* < 0.05 were considered statistically significant, and indicated in figures as follows: * *p* < 0.05; ** *p* < 0.01; *** *p* < 0.001.

## 3. Results

### 3.1. Experimental Results

#### 3.1.1. Measurement of Antiviral Activities of Various Carbazole Derivatives

To screen natural drugs against PEDV, 18 carbazole derivatives (No.1–18) were generated (Figure 1), and added to vero-81 cells prior to EGFP-PEDV infection. Under the fluorescence microscopy observation, EGFP fluorescence was significantly reduced when PEDV-infected cells treated with No.5, No.7, or No.18 carbazole derivatives (Figure 2a). Similarly, qRT-PCR was performed, and it was found that the level of vRNA is reduced to the bottom in No.5, No.7, or No.18 carbazole derivatives treated cells, compared to other carbazole derivatives treated cells (Figure 2b). Consistently, flow cytometry assay also showed that the positive rate of PEDV was significantly reduced in cells incubated with No.5, No.7, or No.18 carbazole derivatives (Figure 3). These results indicated that No.5, No.7, and No.18 carbazole derivatives can significantly inhibit PEDV replication.

#### 3.1.2. No.7 and No.18 Carbazole Derivatives Reveal Low Cytotoxicity in Vero-81 Cells

The cellular cytotoxic effects of No.5, No.7, and No.18 carbazole derivatives on vero-81 cells were determined using the CCK8 kit after cells were treated with No.5, No.7, or No.18 carbazole derivatives for 12 h. Cell viability is downregulated to less than 80% in No.5 carbazole derivative incubated cells, indicating a strong cytotoxicity of No.5 carbazole derivative (Figure 4a). The maximum concentrations (40 µM) of No.7 carbazole derivative treatment resulted in a cell viability higher than 80%, which indicates relatively low cytotoxicity (Figure 4b). As for No.18 carbazole derivative treated cells, viability is always higher over 80% in comparison with control (Figure 4c). In accordance, 10, 20, and 40 µM concentrates of No.7; and 20, 40, and 60 µM concentrates of No.18 carbazole derivative were used in subsequent study.

#### 3.1.3. Both No.7 and No.18 Carbazole Derivatives Inhibit PEDV Proliferation Dose-Dependently

To determine whether No.7 and No.18 carbazole derivatives combating PEDV are dose-dependent or not, vero-81 cells were incubated with increasing concentrations of No.7 (10, 20, 40 µM) or No.18 (20, 40, 60 µM) carbazole derivatives, and then infected with PEDV for 24 h. The supernatants were collected for TCID_50_, and it was found that viral titration significantly declined compared to mock-treated cells (Figure 5a,b). In addition, cells were harvested for qRT-PCR or western blot, which showed that the mRNA (Figure 5c,d) or protein expression levels (Figure 5e,f) of PEDV N decreased in a dose-dependent manner. These results indicate that No.7 and No.18 carbazole derivatives combating PEDV replication did so in a dose-dependent manner.

#### 3.1.4. Time-Dependent Effect of Drug on PEDV Replication

To further illustrate the effect of carbazole derivatives on PEDV infection, we performed time of addition experiments. Vero-81 cells were seeded into 12-cell-plates, and grown up to 80% confluence. Then, cells were infected with PEDV (MOI = 0.1) for 1 h, followed by No.7 or No.18 carbazole derivatives incubation at 1, 3, 6, 9, 12, 15, 18, and 21 hpi (Figure 6a). At 24 hpi, cell culture supernatants were collected for TCID_50_ to determine viral titer. PEDV viral titer was significantly reduced from 1 hpi to 21 hpi in No.7 carbazole derivatives treated cells (Figure 6b), whereas it decreased only at 1 and 3 hpi in No.18 carbazole derivatives treated cells (Figure 6c), indicating that No.7 and No.18 carbazole derivatives had a stronger inhibitory effect during the initial stages of PEDV infection.

#### 3.1.5. No.7 and No.18 Carbazole Derivatives Inhibit PEDV Attachment to Cells

To further explore in which step of the PEDV life cycle the No.7 and No.18 carbazole derivatives function, qRT-PCR was applied to evaluate the effect of drugs on PEDV attachment and entry. As shown in Figure 7a,b, No.7 and No.18 carbazole derivatives treatment led to a robust reduction of viral mRNA levels in comparison with mock-treated cells in viral attachment assay, whereas no significant differences arose in viral entry assay. These data indicated that carbazole derivatives possess anti-PDEV activity at the viral attachment stage.

## 4. Discussion

PED is a worldwide epidemic pig disease caused by PEDV, which seriously harms the breeding industry. Owing to the antibody interference of other vaccines, high frequency mutation of the virus itself, and some other reasons we mentioned earlier, even though there are many vaccines against PED on the market, their effects are limited, which has caused an urgent and pertinent need for the development of new antiviral therapies [31,32]. In this study, it was found, for the first time, that carbazole alkaloid derivatives, types of modified natural product extracts, had anti-PEDV ability, which could play a role in the PEDV attachment stage, and effectively inhibit virus proliferation. The relevant results suggest carbazole derivatives as a potential candidate for PEDV treatment.

Plants and plant-derived compounds are exploited extensively as candidates for new antiviral agents due to few side effects, high availabilities, and low cost [33]. The carbazole alkaloid derivatives used in this study are also plant-derived compounds, which are new compounds synthesized by modifying the original basic structure of a carbazole alkaloid by adding or substituting functional groups. During the 18 carbazole alkaloid derivatives, only No.5, No.7, and No.18 have a significant anti-PEDV effect. Intriguingly, these three compounds present structural characteristics similar to other naturally occurring compounds that demonstrated promising anti-coronavirus activity [34,35,36,37]. All the compounds possess the aromatic rings and substituted fused rings, which indicate that synthetic modifications can be made to possibly increase the compounds’ anti-PEDV activities.

Although No.5, No.7, and No.18 out of the 18 carbazole alkaloid derivatives were found to have anti-PEDV activity through preliminary screening, we did not choose No.5, because it has a certain cytotoxicity which may influence the judgment of the antiviral effect of the drug in vitro. We performed the subsequent experiments with No.7 and No.18, and found that they can inhibit PEDV through preventing viral attachment, but the specific mechanism is not yet known, and needs to be further analyzed.

Some natural products and their derivatives have been approved to be against PEDV, which either act as host-targeting agents, such as immunity or cell survival modulators, or viral-targeting agents, such as spike protein, polymerase, or protease to interfere with the viral replication cycle that includes attachment, entry, transcription and replication, and assembly and release [37,38]. For example, Griffithsin had potent anti-PEDV activity via the prevention of viral attachment and cell-to-cell spread [39]. Pogostemon cablin polysaccharide was reported to possess anti-oxidative abilities, leading to the inhibition of PEDV infection [40]. Puerarin (PR), a major isoflavonoid isolated from the Chinese herb Gegen, possesses many pharmacological activities, including anti-inflammatory and anti-viral activities [41]. PR protects against PEDV infection via regulating the interferon and NF-κB signaling pathway [42]. The carbazole alkaloid derivatives studied in this paper play a role in the attachment stages. However, all drugs reported cannot completely inhibit PEDV; thus, a combination of multiple drugs can be tried to achieve better antiviral effects.

In summary, this study represents an important first report on carbazole derivatives against PEDV. Further studies in live animals will be necessary to develop a novel antiviral strategy for PEDV therapy.

## Figures and Tables

**Figure 1 viruses-13-02527-f001:**
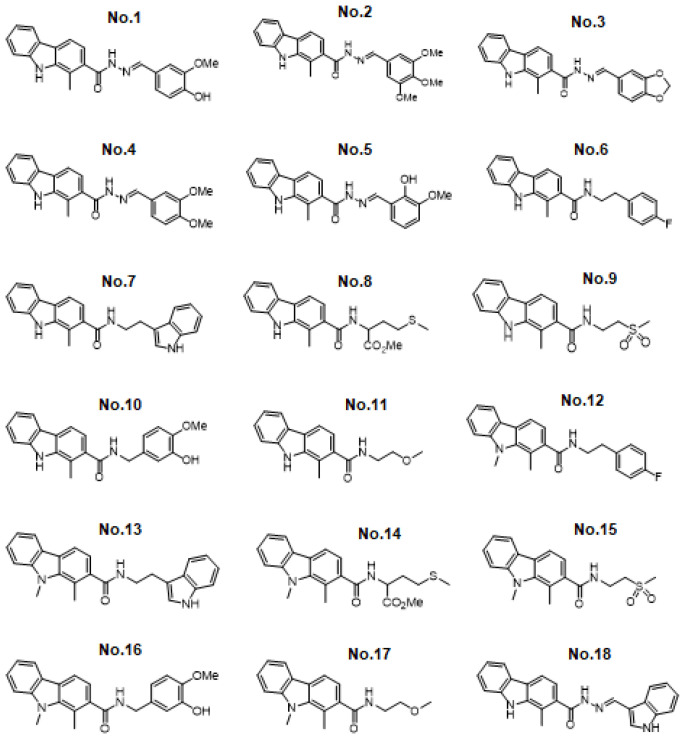
Chemical structures of 18 synthetic carbazole alkaloids.

**Figure 2 viruses-13-02527-f002:**
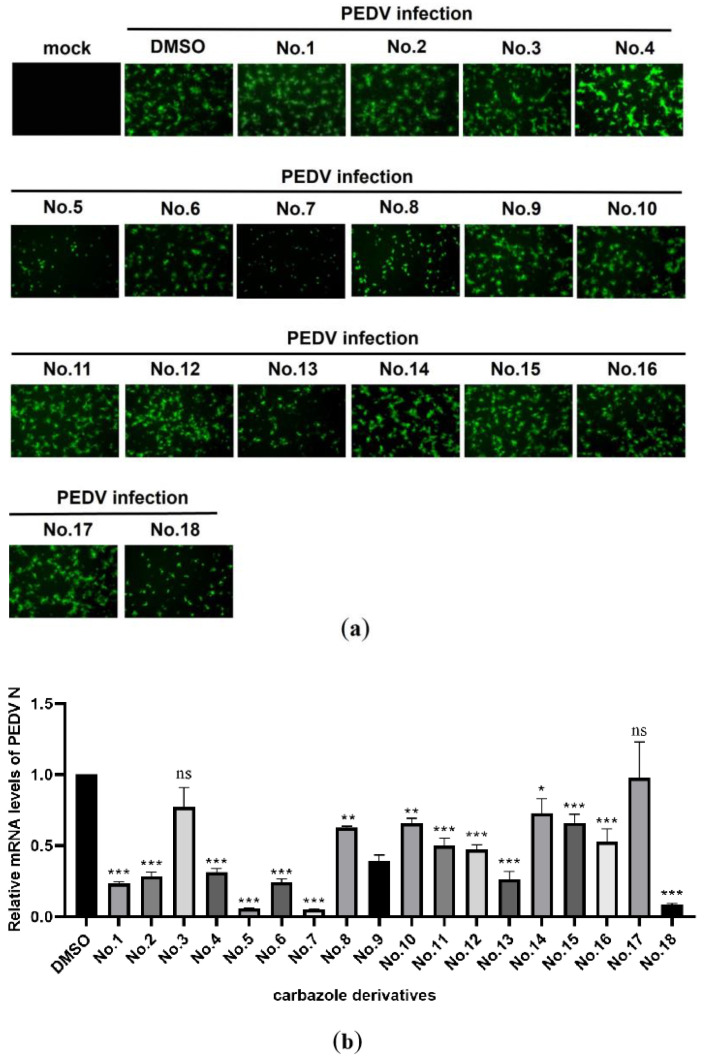
Evaluation anti-PEDV activities of 18 carbazole derivatives. Vero-81 cells were treated with 18 kinds (from No.1 to No.18) of carbazole derivatives, and then infected with EGFP-PEDV for 24 h. (**a**) The fluorescence in the cells were observed by fluorescence microscope. (**b**) RNA was extracted from PEDV-infected or non-infected vero-81 cells. The relative mRNA level of PEDV N was determined by qRT-PCR. * *p* < 0.05; ** *p* < 0.01; *** *p* < 0.001; ns not significant.

**Figure 3 viruses-13-02527-f003:**
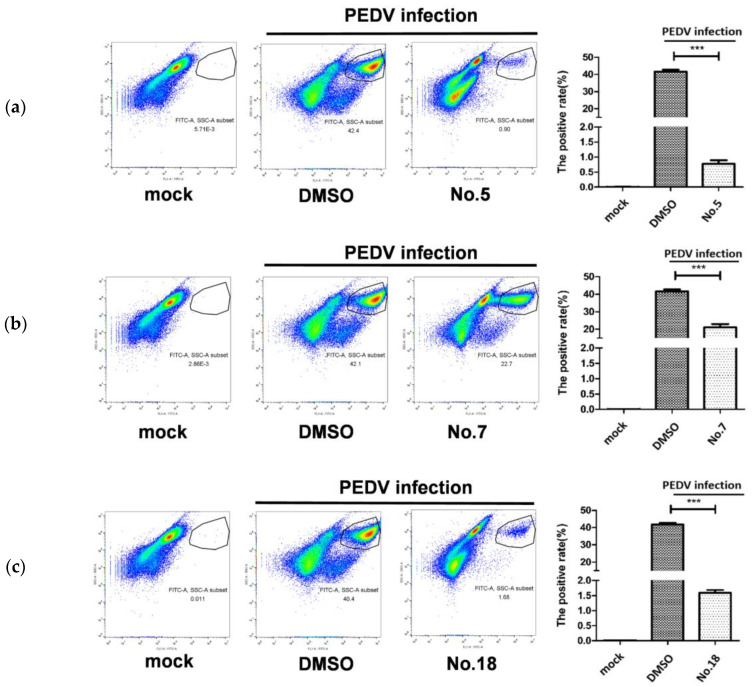
No.5, No.7, and No.18 carbazole derivatives were confirmed to anti-PEDV replication. (**a**–**c**) The vero-81 cells were incubated with No.5 (**a**), No.7 (**b**), or No.18 (**c**) carbazole derivatives at a concentration of 10 µM/mL for 1 h, and then infected EGFP-PEDV for 24 h. The fluorescence was analyzed by flow cytometry. *** *p* < 0.001.

**Figure 4 viruses-13-02527-f004:**
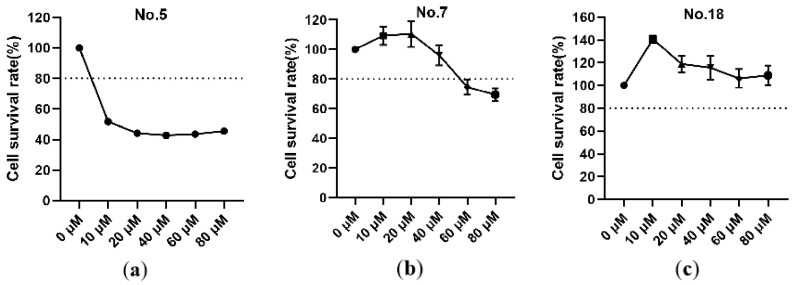
No.7 and No.18 carbazole derivatives had low toxicity to vero-81 cells. (**a**–**c**) Cells were treated with increasing concentration (0, 10, 20, 40, 60, 80 µM) of No.5 (**a**), No.7 (**b**), or No.18 (**c**) carbazole derivatives for 24 h. The relative cell viability was evaluated by the CCK8 Kit. The dotted line indicates the 80% cytostatic concentration.

**Figure 5 viruses-13-02527-f005:**
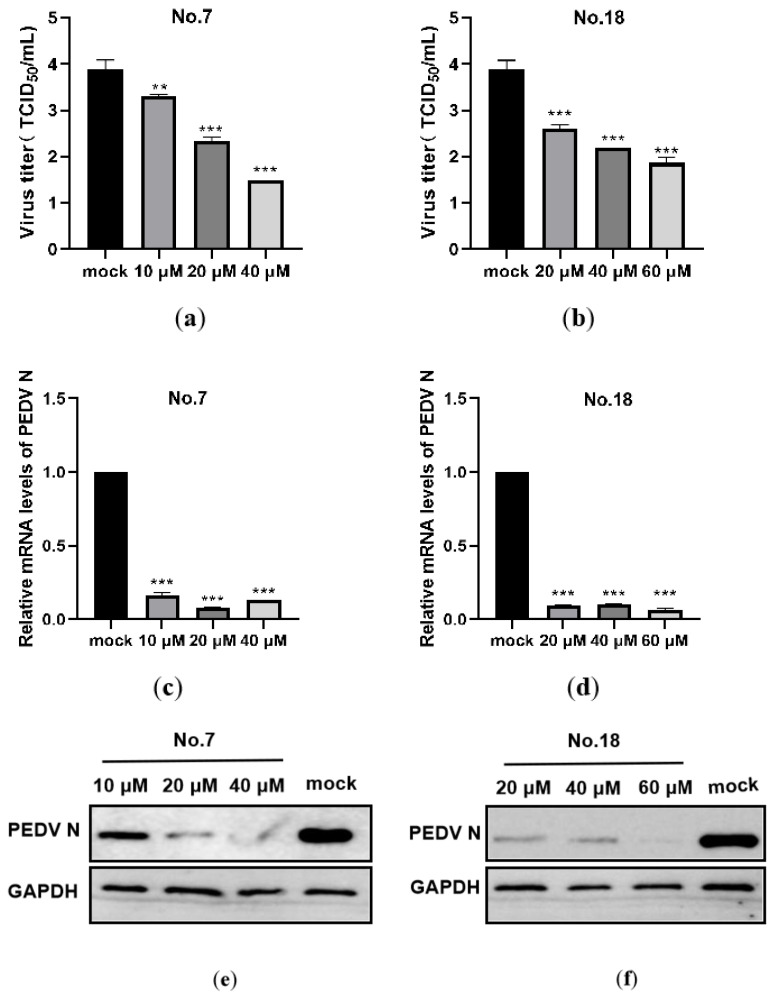
No.7, No.18 carbazole derivatives inhibited PEDV replication were in a dose-dependent manner. The vero-81 cells were incubated with increasing concentrates of carbazole derivatives (10, 20, 40 µM for No.7 (**a**,**c**,**e**); and 20, 40, 60 µM for No.18 (**b**,**d**,**f**)) for 1 h, and then infected with PEDV for 2 h. Then, the cells were treated with the same concentration of carbazole derivatives for 24 h. (**a**,**b**) Cells were collected for RNA extraction, and the relative vRNA level was determined by qRT-PCR. (**c**,**d**) Supernatants were harvested to determine viral titer by TCID_50_. (**e**,**f**) Cells were lysed and subjected to western blot analysis. ** *p* < 0.01; *** *p* < 0.001.

**Figure 6 viruses-13-02527-f006:**
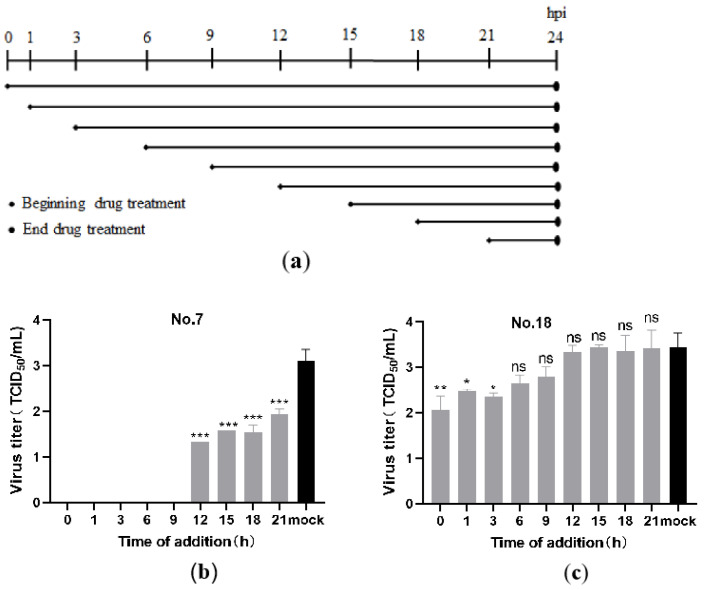
Effects of No.7 and No.18 carbazole derivatives on PEDV replication were time-dependent. (**a**) Schematic representation of carbazole derivatives treatment time. (**b**,**c**) Vero-81 cells were incubated with PEDV (MOI = 0.1) for 1 h, followed by treatment with 40 µM No.7 (**b**) or No.18 (**c**) carbazole derivatives at the indicated time (hpi). The viral titer in cell supernatant was detected by TCID_50_, and calculated by the Reed–Muench method. * *p* < 0.05; ** *p* < 0.01; *** *p* < 0.001.

**Figure 7 viruses-13-02527-f007:**
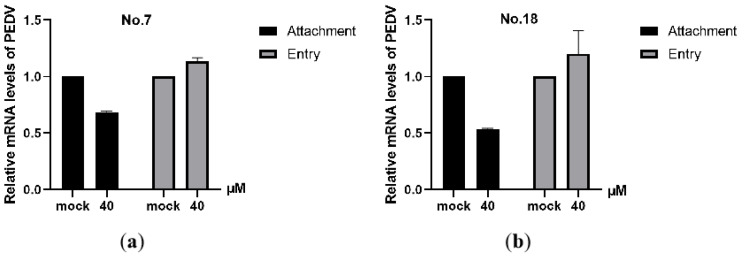
No.7 and No.18 carbazole derivatives inhibit PEDV attachment, but not entry. (**a**,**b**) Levels of vRNA in cells treated with 40 µM No.7 (**a**) or No.18 (**b**) carbazole derivatives during viral attachment and entry were determined by qRT-PCR.

**Table 1 viruses-13-02527-t001:** Primers for quantitative real-time PCR.

Name	5′→3′
qGAPDH-F	5′-CTGCCGTCTGGAAAAACCTG-3′
qGAPDH-R	5′-CGTCGAAGGTGGAAGAGTGG-3′
qPEDV-F	5′-GAGGGTGTTTTCTGGGTTG-3′
qPEDV-R	5′-CGTGAAGTAGGAGGTGTGTTAG-3′

## Data Availability

Not applicable.

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
