# Peer review of "Antiviral Activities of Carbazole Derivatives against Porcine Epidemic Diarrhea Virus In Vitro"

_viruses, 2021, doi:10.3390/v13122527_

Round 1
Reviewer 1 Report
In the manuscript entitled " Antiviral activities of carbazole derivatives against porcine ep-idemic diarrhea virus in vitro", the authors synthesized 18 carbazole alkaloid derivatives and investigated their effects on PEDV replication. The work identified No.7 and No.18 carbazole derivatives can inhibit PEDV with low cell toxicity and in a dose-dependent manner. Subsequently, they illustrated that both drugs can inhibit the early stage of the viral life cycle. The experimental approach was straight forward and logical, while the writing and content need to be further improved.
Suggestions:
1. Some descriptions in the material and method part are too colloquial and wordy, the quality of English needs improving.
- What’s the effect of No.7 and No.18 carbazole derivatives on other enteric coronavirus such as PDCoV? The authors can perform some in vitro experiments of the drugs on PDCoV since these two viruses are very similar, so as to make the content of the manuscript more substantial.
Author Response
- Some descriptions in the material and method part are too colloquial and wordy, the quality of English needs improving.
Response: Languages were minor modified in revised manuscript.
- What’s the effect of No.7 and No.18 carbazole derivatives on other enteric coronavirus such as PDCoV? The authors can perform some in vitro experiments of the drugs on PDCoV since these two viruses are very similar, so as to make the content of the manuscript more substantial.
Response: According to the comments, RT-qPCR was performed to assess the effects of No.7 and No.18 carbazole derivatives on PDCoV replication. As shown below, both No.7 and No.18 carbazole derivatives can also inhibit PDCoV replication. But the data was part of another paper, we did not include in this paper.

Reviewer 2 Report
Dear Authors, please correct the marked words!

Author Response
please correct the marked words!
Response: We apologized for word mistakes, and marked words were corrected.